# Factors affecting anxiety among administrative officers working within the urgent protective action planning zone of a nuclear power station

Hiroko Hori[1], Makiko Orita[1]*, Yasuyuki Taira[1], Hitomi Matsunaga[1], Takashi Kudo[2], Noboru Takamura[1]

1 Department of Global Health, Medicine and Welfare, Nagasaki University Graduate School of Biomedical Sciences, Nagasaki, Japan, 2 Department of Radioisotope Medicine, Nagasaki University Graduate School of Biomedical Sciences, Nagasaki, Japan

* orita@nagasaki-u.ac.jp

## Abstract

The aim of this study was to clarify the factors affecting anxiety among administrative officers working within the urgent protective action planning zone of a nuclear power station to establish an effective education program on radiation and its health effects to help reduce anxiety in residents. We included 1,181 officers who worked at local authorities within the urgent protective action planning zone of Sendai Nuclear Power Station in Kagoshima Prefecture, Japan. Logistic regression analysis revealed that female sex (odds ratio = 2.33), working more than 21 years as an administrative officer (odds ratio = 1.49), lack of participation in training on nuclear disasters (odds ratio = 1.42), and not knowing the three principles of radiation protection (odds ratio = 1.36) were independently associated with anxiety among administrative officers working within the urgent protective action planning zone. It is important to establish an effective education program on radiation and its health effects for administrative officers working within the urgent protective action planning zone to reduce anxiety in residents.

## Introduction

Extensive damage was caused to Tokyo Electric Power Company's Fukushima Daiichi Nuclear Power Plant (FDNPS) as a result of the Great East Japan Earthquake and resulting tsunami that occurred on March 11, 2011. Accordingly, residents near the FDNPS were evacuated within a few days and foodstuffs were controlled within 1 or 2 weeks. External doses in Fukushima City determined by personal dosimeters were 1 mSv/3 months (September–November 2011) in 99.7% of residents (maximum: 2.7 mSv) [1, 2]. Thyroid radiation dose was 10 mSv (maximum: 35 mSv) in 95.7% of children [3]. These measurements of external and internal radiation exposure of residents surrounding the FDNPS have been reported by several research institutions; they all suggest that external and internal radiation doses caused by the

**Funding:** The authors received no specific funding for this work.

**Competing interests:** The authors have declared that no competing interests exist.

accident were relatively low and far from any direct health consequences in the general population [3].

Various information was transmitted to the public regarding the radiation-related health risk following the accident and many people in Fukushima felt anxiety about the health effects of radiation exposure, which resulted in social panic. The Fukushima Health Management Survey revealed that among evacuees of the Fukushima disaster, psychological distress was more frequent among people who perceived health effects of radiation exposure to be very likely, it appears that psychological status was related to the perception of radiation risks [4]. This result suggests that incorrect understanding of health effects of radiation may be related to psychological distress [5]. Furthermore, this anxiety was felt even in communities well outside of Fukushima [6].

We previously revealed that employees of ordinary companies and nurses at Fukushima Medical University Hospital intended to leave their jobs during the radiation emergency following the accident and that the lack of knowledge about radiation was behind it [7, 8]. The Investigation Committee on the Accident at the Fukushima Nuclear Power Station suggested that "there should be many opportunities for citizens to deepen their knowledge and understanding of radiation", and that junior high school science classes and medical education should be provided to help residents acquire knowledge about radiation and its potential health effects [9].

In October 2012, the Nuclear Regulation Authority of Japan established the "Nuclear Emergency Response Guidelines" to provide information on dealing with various situations on the assumption that safety measures could prevent serious accidents. The guidelines recommend establishing priority areas where preventive protective measures are taken based on the state of the facility before the release of radioactive material. The precautionary action zone (PAZ) is an area where evacuation is conducted proactively within a radius of 5 km from a nuclear power plant before the release of radioactive materials. The urgent protective action planning zone (UPZ) is an area located within a radius of 30 km outside the PAZ, where preventive protection measures including sheltering in place, evacuation and temporary relocation are carried out in stages [10]. Local governments where nuclear power plants are located should prepare a nuclear disaster prevention plan, and residents in these areas should participate in nuclear disaster prevention drills. Further, in the event of a radiological emergency, the local government is responsible for providing potassium iodide supplements to the population [10].

To more effectively promote nuclear emergency preparedness in the future, and to inform the correct information on the nuclear emergency preparedness to the public, anxieties need to be eliminated by spreading accurate information about radiation among administrative officers in areas where nuclear power plants are located.

The aim of this study was to clarify the factors related to anxiety during work (ADW) among administrative officers working within one UPZ with the goal of establishing an effective education program on radiation and its health effects to help reduce anxiety in residents.

## Materials and methods

### Study participants

This study was conducted in September 2015. We distributed questionnaires to all 1,558 general administrative officers who worked at local public offices within the UPZ of Sendai Nuclear Power Station (SNPS) in Kagoshima Prefecture (Fig 1), which was the first nuclear power station restarted after the FDNPS accident in Japan. The 1,181 (75.8%) officers who provided complete responses to the questionnaires were included in the analysis.

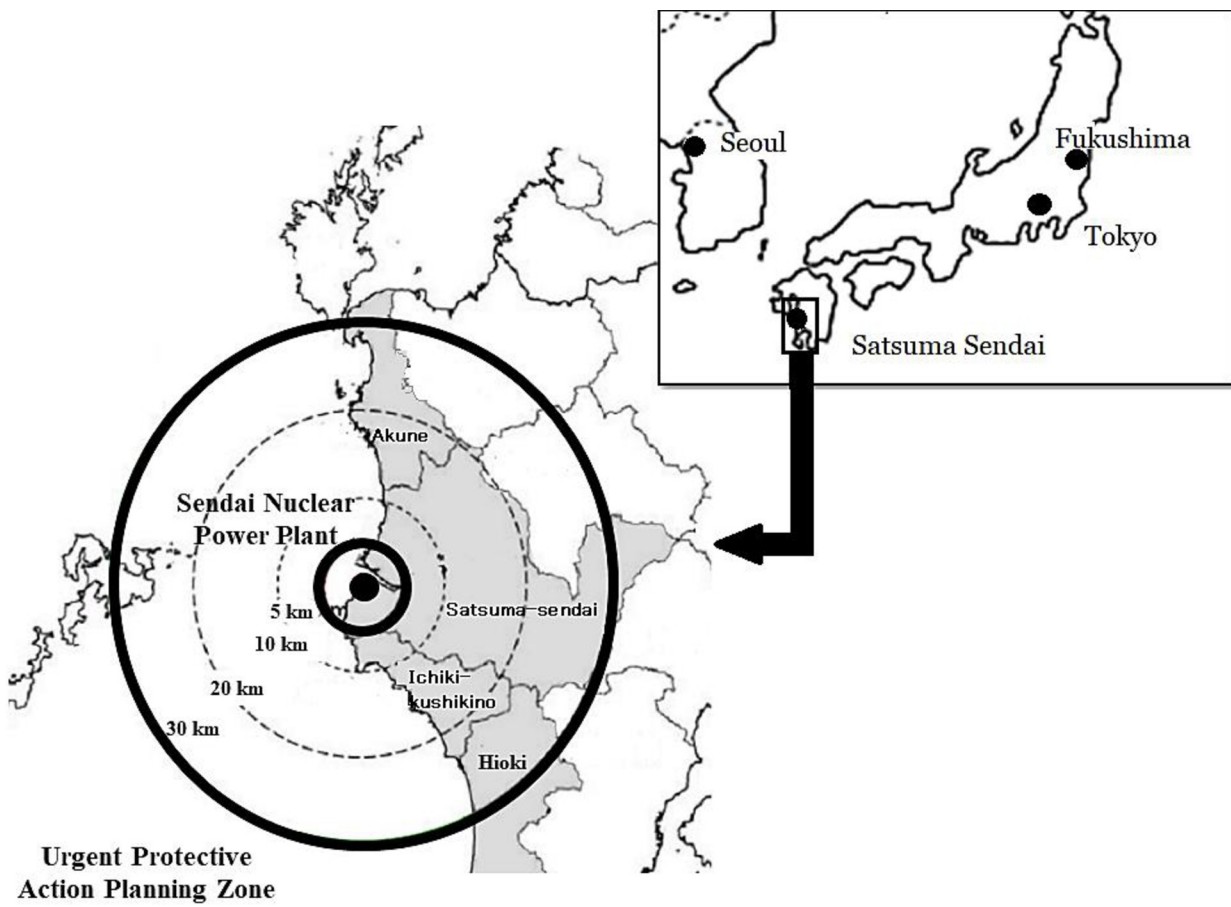

**Fig 1. Location of Sendai Nuclear Power Station, Kagoshima Prefecture, Japan.**

This study was approved by the ethics committee of Nagasaki University Graduate School of Biomedical Sciences (No. 1703021) and Kagoshima University Graduate School of Health Sciences (No. 324).

## Questionnaire

The questionnaire used in this study was developed based on questionnaires used in previous studies conducted in Fukushima Prefecture on intention to leave work [7, 8] and on the mental health and risk perception survey within the framework of the Fukushima Health Management Survey, which was organized by Fukushima Prefecture [4]. The questionnaire was verified with the confirmation of the local government office and Kagoshima University Graduate School of Health Sciences (No. 324). The original questionnaire form is provided in S1 and S2 Files.

In the questionnaire, we asked about the factors associated with ADW and about their level of knowledge of radiation, including the three principles of radiation protection. We also asked questions on the following: demographic factors including sex, age, number of family members, whether they had children under 15 years of age, whether they owned their house, the distance between their house and the SNPS, number of years living at the current address, number of years working, official position, type of work, number of years working in the current section, their level of knowledge about the regional plan for nuclear disaster prevention,

their level of knowledge of their own role in the regional plan for nuclear disaster prevention, experience consulting citizens about radiation, whether they had participated in training on nuclear disasters and whether they were hesitant to be exposed to medical radiation.

## Statistical analysis

We defined the "ADW (+) group" as "officers who were presence of anxiety during work" and the "ADW (−) group" as "officers who were absence of anxiety during work." We identified the factors associated with ADW using the chi-square test. We then used logistic regression analysis and calculated the odds ratios (OR) to identify the factors independently associated with ADW. The original dataset is provided in S1 Dataset. *P*-values less than 0.05 were considered significant. Statistical analyses were performed using IBM SPSS Statistics 23 software (Armonk, NY).

## Results

Among the 1,181 administrative officers that participated in the study, 640 (54.2%) had ADW, and 178 (15.1%) responded that citizens had asked them questions about radiation. Female officers had significantly higher anxiety than male officers (27.5% vs. 12.9%, p<0.001). Officers who were living within 20 km of the SNPS (Table 1), who were hesitant to be exposed to medical radiation (Table 2), who had not participated in training for nuclear disasters (Table 2) and who did not know the three principles of radiation protection (Table 3), had significantly higher anxiety.

Logistic regression analysis revealed that female sex (OR = 2.33, 95% confidence interval [CI]: 1.70–3.20, p<0.001), working more than 21 years as an administrative officer (OR = 1.49, 95% CI: 1.17–1.90, p = 0.01), lack of participation in training on nuclear disasters (OR = 1.42, 95% CI: 1.08–1.87, p = 0.012), and not knowing the three principles of radiation protection (OR = 1.36, 95% CI: 1.04–1.78, p = 0.026) were independently associated with having ADW (Table 4).

## Discussion

Over 9 years have passed since the accident at the FDNPS. In Fukushima Prefecture, the recovery of municipalities such as Kawauchi Village, Naraha Town, and Tomioka Town is

**Table 1. ADW among administrative officers within the UPZ.**

| Item | ADW (+) n = 640 (%) | ADW (-) n = 541 (%) | p |
|---|---|---|---|
| Female sex | 176 (27.5) | 70 (12.9) | <0.001 |
| Older than 40 years of age | 440 (68.8) | 359 (66.4) | 0.381 |
| Single household | 57 (8.9) | 62 (11.5) | 0.146 |
| Living with children younger than 15 years old | 286 (44.7) | 237 (43.8) | 0.762 |
| Living in their own house | 455 (71.1) | 392 (72.5) | 0.604 |
| Living within 20 km of the nuclear power plant | 320 (50.0) | 338 (62.5) | <0.001 |
| Living in the area for more than 20 years | 255 (39.8) | 47 (24.1) | <0.001 |
| Hesitant to be exposed to medical radiation | 146 (22.8) | 61 (11.3) | <0.001 |

Numbers indicate survey participants who responded "yes." Percentages indicate the proportion of participants who responded "yes."

ADW: anxiety during work; UPZ: urgent protective action planning zone; ADW (+): presence of anxiety during work; ADW (-): absence of anxiety during work.

**Table 2. Working situation by ADW among local administrative officers within the UPZ.**

| Item | ADW (+) n = 640 (%) | ADW (-) n = 541 (%) | p-value |
|---|---|---|---|
| Working more than 21 years | 368 (57.5) | 270 (49.9) | 0.009 |
| Working as general staff | 368 (57.5) | 301 (55.6) | 0.520 |
| Working as staff of radiation-related institutions | 109 (17.0) | 94 (17.4) | 0.876 |
| Experience being asked about radiation by residents | 92 (14.4) | 86 (15.9) | 0.467 |
| Having knowledge of the regional plan for nuclear disaster prevention | 367 (57.3) | 347 (64.1) | 0.017 |
| Having knowledge of their role in the regional plan for nuclear disaster prevention | 282 (44.1) | 312 (57.7) | <0.001 |
| No experiences of training on nuclear disaster. | 350 (54.7) | 208 (38.4) | <0.001 |

Numbers indicate survey participants who responded "yes." Percentages indicate the proportion of participants who responded "yes."
ADW: anxiety during work; UPZ: urgent protective action planning zone; ADW (+): presence of anxiety during work; ADW (-): absence of anxiety during work.

progressing [11]. In regions of Japan where nuclear power plants are located, nuclear disaster prevention drills are being organized by prefectural and national governments to prepare for possible nuclear accidents in the future.

In this study, we identified the factors related to ADW among administrative officers working within the UPZ of the SNPS, which was the first nuclear power station to restarted after the FDNPS accident. Our results revealed that 54.2% of administrative officers were concerned about the health effects of radiation exposure while working within the UPZ. The United Nations Scientific Committee on the Effects of Atomic Radiation reported that there would be no detectable direct health effects of radiation exposure on the general public following the FDNPS accident [12]. Nevertheless, administrative officers working in the UPZ of the SNPS were worried about the health effects of radiation.

In our study, we showed that sex, number of years working, participation in training on nuclear disasters and perceptions of the health effects of radiation were independently associated with administrative officers' anxiety about the health effects of radiation exposure while working within the UPZ of the SNPS.

**Table 3. Knowledge of radiation by ADW among local authorities within the UPZ.**

| Item | ADW (+) n = 640(%) | ADW (-) n = 541(%) | p-value |
|---|---|---|---|
| Not knowing the three principles of radiation protection | 455 (71.1) | 316 (58.4) | <0.001 |
| Not knowing the types of radiation | 223 (34.8) | 157 (29.0) | 0.033 |
| Not knowing the units of measuring radiation | 412 (64.4) | 274 (50.6) | <0.001 |
| Not knowing about half-decay of radionuclides | 291 (45.5) | 186 (34.4) | <0.001 |
| Not knowing about internal and external radiation exposure | 186 (29.1) | 147 (24.4) | 0.072 |
| Not knowing the annual radiation dose limit for the public | 215 (33.6) | 156 (28.8) | 0.079 |
| Not knowing about stable potassium iodide | 92 (14.4) | 80 (14.8) | 0.841 |

ADW: anxiety during work; UPZ: urgent protective action planning zone; ADW (+): presence of anxiety during work; ADW (-): absence of anxiety during work.

**Table 4. Logistic regression analysis for ADW of local authorities within the UPZ.**

| Variables | Item | OR | 95% CI | p-value |
|---|---|---|---|---|
| Sex | Female/Male | 2.33 | 1.70–3.20 | <0.001 |
| Years working as an administrative officer | 21 years or more/Less than 21 years | 1.49 | 1.17–1.90 | 0.001 |
| Participation in training on nuclear disasters | 0 times/More than 1 time | 1.42 | 1.08–1.87 | 0.012 |
| Three principles of radiation protection | Unknown/know | 1.36 | 1.04–1.78 | 0.026 |
| Distance from house to the nuclear power plant | More than 20 km/Within 20 km | 1.30 | 0.99–1.68 | 0.052 |

ADW: anxiety during work; UPZ: urgent protective action planning zone; OR: odds ratio; 95% CI: 95% confidence interval.

We indicated that females expressed greater concern about the health effects of radiation exposure than males, which is consistent with previous studies [7, 8]. While no qualitative exploration was done in this particular instance, it may be surmised that women, especially those of childbearing age, may be particularly concerned about the possible effects of radiation on fertility and progeny [7, 8, 13, 14]. We also found that number of years working as an administrative officer was significantly associated with ADW within the UPZ. Previously, we reported that nurses' intention to leave their job in Fukushima after the FDNPS accident was significantly associated with younger age and shorter working tenure [7, 15, 16]. On the other hand, we also surveyed non-medical employees who had been working in areas surrounding the FDNPS and revealed that age and working tenure were not significantly different between employees who intended to leave their job and those who did not [8, 17]. Such inconsistent results may be due to occupational responsibilities during the nuclear disaster.

In addition, our present results showed that place of work, participation in training on nuclear disasters, and knowledge of the three principles of radiation protection were independently associated with administrative officers' anxiety regarding the health effects of radiation exposure due to working within the UPZ [18]. It was reported that administrative officers in Fukushima who are responsible for guiding citizens during the recovery process did not have enough knowledge about the health effects of radiation, which caused delays in the recovery of the community [19]. It was also pointed out that insufficient unity and lack of credibility of administrative officers and teachers caused anxiety within the public [19]. Furthermore, the International Commission on Radiological Protection emphasized the need for dissemination of radiation protection knowledge to the public and stated that the key to disaster recovery is for professionals responsible for public health and education to create a culture of national radiation protection [20]. In Japan, the Nuclear Regulatory Agency and other ministries and agencies have created training facilities and programs on radiation and nuclear disaster response. Administrative officers who are responsible for responding to the public should participate in such training to ensure effective communication. Also, it is necessary for administrative officers to build their credibility with the public through daily communication, and to disseminate accurate information about radiation [21]. From this point of view, it is important for administrative officers to develop a comprehensive educational program to build trust with the public and reduce public concerns about radiation exposure and its health effects with the cooperation of radiation medical science experts.

## Study limitations

The present study had several limitations. First, there might have been a selection bias since the study was conducted only in the UPZ of the SNPS. Therefore, the generalizability of the results is limited. Second, in the questionnaire, we could not obtain sufficient information on potential confounding factors, such as detailed lifestyle habits, educational environment, the

degradation in trust in the central government after the Fukushima disasters and the responsibilities of study participants related to nuclear disasters.

## Conclusion

We revealed that sex, number of years working, participation in training on nuclear disasters, and not knowing the three principles of radiation protection were independently associated with ADW among administrative officers within one UPZ. It is important to establish an effective education program on radiation and its health effects for administrative officers working within the UPZ to reduce the anxiety of residents.

## Supporting information

**S1 File. The original questionnaire form in Japanese.**
(DOCX)

**S2 File. The original questionnaire form translated into English.**
(DOCX)

**S1 Dataset. The original dataset of this study.**
(XLSX)

## Author Contributions

**Conceptualization:** Hiroko Hori, Noboru Takamura.

**Data curation:** Hiroko Hori.

**Formal analysis:** Hiroko Hori, Makiko Orita, Noboru Takamura.

**Investigation:** Hiroko Hori, Noboru Takamura.

**Methodology:** Takashi Kudo.

**Writing – original draft:** Hiroko Hori, Makiko Orita, Takashi Kudo, Noboru Takamura.

**Writing – review & editing:** Makiko Orita, Yasuyuki Taira, Hitomi Matsunaga, Takashi Kudo, Noboru Takamura.

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
