## [Decision Letter · Decision Letter 0]

15 Jun 2020

PONE-D-20-01520

Factors affecting anxiety among administrative officers working within the urgent protective action planning zone of a nuclear power station

PLOS ONE

Dear Dr. Orita,

Thank you for submitting your manuscript to PLOS ONE. After careful consideration, we feel that it has merit but does not fully meet PLOS ONE’s publication criteria as it currently stands. Therefore, we invite you to submit a revised version of the manuscript that addresses the points raised during the review process.

I appreciate the amount of work the authors have put into this study, which is an important one given the lacunae in this area. There are a few minor issues to be addressed in this manuscript moving forward: in particular I hope the authors put extra focus on responding to Reviewer 1's comments on the somewhat problematic usage of "maternal characteristics of women" that appears throughout the manuscript. While I think many individuals may appreciate the innocuous sentiment behind the statement, the authors need to rephrase this or qualify the statement: are they referring to the societal duties imposed upon women, etc.? 

We look forward to receiving your revised manuscript.

Kind regards,

Haikel A. Lim, MD, MSc

Academic Editor

PLOS ONE

Journal Requirements:

2. Please include additional information regarding the survey or questionnaire used in the study and ensure that you have provided sufficient details that others could replicate the analyses. For instance, if you developed a questionnaire as part of this study and it is not under a copyright more restrictive than CC-BY, please include a copy, in both the original language and English, as Supporting Information. In addition, please provide any details of the pre-testing of this questionnaire that took place.

Additional Editor Comments (if provided):

I appreciate the amount of work the authors have put into this study, which is an important one given the lacunae in this area. There are a few minor issues to be addressed in this manuscript moving forward: in particular I hope the authors put extra focus on responding to Reviewer 1's comments on the somewhat problematic usage of "maternal characteristics of women" that appears throughout the manuscript. While I think many individuals may appreciate the innocuous sentiment behind the statement, the authors need to rephrase this or qualify the statement: are they referring to the societal duties imposed upon women, etc.?

Reviewers' comments:

Reviewer's Responses to Questions

**Comments to the Author**

1. Is the manuscript technically sound, and do the data support the conclusions?

Reviewer #1: Yes

Reviewer #2: Yes

Reviewer #3: Yes

2. Has the statistical analysis been performed appropriately and rigorously? 

Reviewer #1: I Don't Know

Reviewer #2: Yes

Reviewer #3: Yes

3. Have the authors made all data underlying the findings in their manuscript fully available?

Reviewer #1: Yes

Reviewer #2: Yes

Reviewer #3: Yes

4. Is the manuscript presented in an intelligible fashion and written in standard English?

Reviewer #1: Yes

Reviewer #2: Yes

Reviewer #3: Yes

5. Review Comments to the Author

Reviewer #1: The article is an important contribution to the field. A few suggested edits are below:

- The background primarily focuses on radiation exposure, but the article is focused on risk perception. Revising this section to highlight some data on risk perception would be more congruent with the paper.

- The authors speculation on causes of significant findings requires more information. Particularly line 164 discussing the potential causal factor being "the maternal characteristics of women". This needs more explanation. The citations are also focused on burnout of nurses in the context of occupational stress and burnout.

- The conclusions to build a comprehensive education program should take into account the degradation in trust in government authorities after the 3/11 disasters. Some acknowledgement of the research in this regard could also be cited in the conclusions and/or in the study limitations.

Reviewer #2: The paper had been able to comply with the standards of the Journal. However, some few points needs to be define further.

1. How did they validate the questionnaires used in this study?

2. For the population sampling, how did you choose your study subjects?

3. Some few grammatical errors had been observed.

Reviewer #3: There are not many studies on radiation anxiety related with nuclear plant disaster in Fukushima. This is a good research with technical soundness, appropriate statistical analysis, sharing of relevant data in the paper and written in standard English.

6. PLOS authors have the option to publish the peer review history of their article (what does this mean?). If published, this will include your full peer review and any attached files.

Reviewer #1: No

Reviewer #2: No

Reviewer #3: No

---

## [Author Response · Author response to Decision Letter 0]

26 Jun 2020

Dear Editor:

We greatly appreciate your comments on our manuscript and provide point-by-point responses to the issues raised by the reviewers below. We have revised some sections of the text and added more information. In particular, according to your suggestion, we revised the text that reviewer 1 commented on as follows: “We indicated that females expressed greater concern about the health effects of radiation exposure than males, which is consistent with previous studies [7, 8]. This might be due to women’s motherly instincts, such as feeling anxiety about childbearing and raising children [7, 8, 13, 14]” (Page 14, Lines 175 – 178).

Response to Journal Requirements:

We greatly appreciate your comments on our manuscript. We have revised the manuscript according to your suggestions. We adjusted our formatting to ensure that our manuscript meets PLOS ONE’s style requirements including adding the captions for the Supporting information files at the end of the manuscript (Title page, Reference and Page 19, Lines 314 – 320).

We added a copy of the questionnaire used in this study, in both the original language (Japanese) and English, as Supporting information Files S1 and S2. We added the following text in the Materials and methods section: “The questionnaire used in this study was developed based on questionnaires used in previous studies conducted in Fukushima Prefecture on intention to leave work [7, 8] and on the mental health and risk perception survey within the framework of the Fukushima Health Management Survey, which was organized by Fukushima Prefecture [4]. The questionnaire was verified with the confirmation of the local government office and Kagoshima University Graduate School of Health Sciences (No. 324). The original questionnaire form is provided in Supporting information File S1 and File S2.” (Page 5, Lines 98 – 104.)

We added the captions for the Supporting information files at the end of the manuscript and updated references to these files in the text accordingly. (Page 5, Lines 103 – 104; Page 6, Lines 122 – 123; Page 19, Lines 314 – 320.)

Response to Reviewer.

Reviewer #1:

We greatly appreciate your comments on our manuscript. We have revised the manuscript according to your suggestions.

- The background primarily focuses on radiation exposure, but the article is focused on risk perception. Revising this section to highlight some data on risk perception would be more congruent with the paper.

According to your suggestions, we revised the Introduction section. The description of radiation exposure is summarized as follows: “These measurements of external and internal radiation exposure of residents surrounding the FDNPS have been reported by several research institutions; they all suggest that external and internal radiation doses caused by the accident were relatively low and far from any direct health consequences in the general population [3].” (Page 3, Lines 38 – 41.)

Text describing individual risk perception after the Fukushima accident and the relevant citations were added as follows: “Various information was transmitted to the public regarding the radiation-related health risk following the accident and many people in Fukushima felt anxiety about the health effects of radiation exposure, which resulted in social panic. The Fukushima Health Management Survey revealed that among evacuees of the Fukushima disaster, psychological distress was more frequent among people who perceived health effects of radiation exposure to be very likely, it appears that psychological status was related to the perception of radiation risks [4]. This result suggests that incorrect understanding of health effects of radiation may be related to psychological distress [5]. Furthermore, this anxiety was felt even in communities well outside of Fukushima [6].” (Page 3, Lines 42 – 50.)

- The authors speculation on causes of significant findings requires more information. Particularly line 164 discussing the potential causal factor being "the maternal characteristics of women". This needs more explanation. The citations are also focused on burnout of nurses in the context of occupational stress and burnout.

According to your suggestion, we revised the text as follows: “We indicated that females expressed greater concern about the health effects of radiation exposure than males, which is consistent with previous studies [7, 8]. This might be due to women’s motherly instincts, such as anxiety about childbearing and raising children [7, 8, 13, 14].” (Page 14, Lines 175 – 178).

- The conclusions to build a comprehensive education program should take into account the degradation in trust in government authorities after the 3/11 disasters. Some acknowledgement of the research in this regard could also be cited in the conclusions and/or in the study limitations.

According to your suggestion, we added the following to the conclusion section: “Also, it is necessary for administrative officers to build their credibility with the public through daily communication, and to disseminate accurate information about radiation [21]. From this point of view, it is important for administrative officers to develop a comprehensive educational program to build trust with the public and reduce public concerns about radiation exposure and its health effects with the cooperation of radiation medical science experts.” (Page 15, Lines 201 – 207). In addition, we have also added the following sentence as a study limitation: “…we could not obtain sufficient information on potential confounding factors, such as detailed lifestyle habits, educational environment, the degradation in trust in the central government after the Fukushima disasters and the responsibilities of study participants related to nuclear disasters.” (Page 15, Lines 212 – 216).

 

Reviewer #2:

We greatly appreciate your comments on our manuscript. We have revised the manuscript according to your suggestions.

1. How did they validate the questionnaires used in this study?

According to your suggestion, we revised the Materials and Methods section as follows: “The questionnaire used in this study was developed based on questionnaires used in previous studies conducted in Fukushima Prefecture on intention to leave work [7, 8] and on the mental health and risk perception survey within the framework of the Fukushima Health Management Survey, which was organized by Fukushima Prefecture [4]. The questionnaire was verified with the confirmation of the local government office and Kagoshima University Graduate School of Health Sciences (No. 324). The original questionnaire form is provided in Supporting information File S1 and File S2.” (Page 5, Lines 98 – 104.)

2. For the population sampling, how did you choose your study subjects?

According to your suggestion, we revised the Materials and Methods section as follows: “We distributed questionnaires to all 1,558 general administrative officers who worked at local public offices within the UPZ of Sendai Nuclear Power Station (SNPS) in Kagoshima Prefecture (Fig 1), which was the first nuclear power station restarted after the FDNPS accident in Japan. The 1,181 (75.8%) officers who provided complete responses to the questionnaires were included in the analysis.” (Page 4 and 5, Lines 84 – 89.)

3. Some few grammatical errors had been observed.

According to your suggestion, a native English speaker checked the grammar in the revised manuscript.

 

Reviewer #3:

We greatly appreciate your comments on our manuscript.

---

## [Editor Report · Decision Letter 1]

29 Jun 2020

PONE-D-20-01520R1

Factors affecting anxiety among administrative officers working within the urgent protective action planning zone of a nuclear power station

PLOS ONE

Dear Dr. Orita,

Thank you for submitting your manuscript to PLOS ONE. After careful consideration, we feel that it has merit but does not fully meet PLOS ONE’s publication criteria as it currently stands. Therefore, we invite you to submit a revised version of the manuscript that addresses the points raised during the review process.

I thank the authors for their revised submission, which, based on my review, has adequately addressed the reviewers concern bar one.

The sentence "This might be due to women’s motherly instincts, such as anxiety about childbearing and raising children [7, 8, 13, 14]" (lines 182-184) is problematic, primarily because of the use of "motherly instincts". I think the authors are trying to suggest that the concerns women might have are not a result of motherly or maternal instincts, but more on their future ability to bear healthy children. 

As such, the sentence may be better explained with something along the lines of: "While no qualitative exploration was done in this particular instance, it may be surmised that women, especially those of childbearing age, may be particularly concerned about the possible effects of radiation on fertility and progeny [7, 8, 13, 14]". 

If the authors are insistent on the use of "motherly instincts", then this discussion section will have to be expanded in order for the authors to adequately support their claim (through quantitative or qualitative studies) that health concerns are a result of female sex.

We look forward to receiving your revised manuscript.

Kind regards,

Haikel A. Lim, MD, MSc

Academic Editor

PLOS ONE

Additional Editor Comments (if provided):

I thank the authors for their revised submission, which, based on my review, has adequately addressed the reviewers concern bar one.

The sentence "This might be due to women’s motherly instincts, such as anxiety about childbearing and raising children [7, 8, 13, 14]" (lines 182-184) is problematic, primarily because of the use of "motherly instincts". I think the authors are trying to suggest that the concerns women might have are not a result of motherly or maternal instincts, but more on their future ability to bear healthy children. As such, the sentence may be better explained with something along the lines of: "While no qualitative exploration was done in this particular instance, it may be surmised that women, especially those of childbearing age, may be particularly concerned about the possible effects of radiation on fertility and progeny [7, 8, 13, 14]". If the authors are insistent on the use of "motherly instincts", then this discussion section will have to be expanded in order for the authors to adequately support their claim (through quantitative or qualitative studies) that health concerns are a result of female sex.

---

## [Author Response · Author response to Decision Letter 1]

30 Jun 2020

We greatly appreciate your comments on our manuscript. According to your suggestion, we changed the applicable part as follows: from “This might be due to women’s motherly instincts, such as anxiety about childbearing and raising children” to “While no qualitative exploration was done in this particular instance, it may be surmised that women, especially those of childbearing age, may be particularly concerned about the possible effects of radiation on fertility and progeny”. (Page 14, Lines 176 – 179). 

Your kind support would be most appreciated.

---

## [Editor Report · Decision Letter 2]

20 Jul 2020

Factors affecting anxiety among administrative officers working within the urgent protective action planning zone of a nuclear power station

PONE-D-20-01520R2

Dear Dr. Orita,

We’re pleased to inform you that your manuscript has been judged scientifically suitable for publication and will be formally accepted for publication once it meets all outstanding technical requirements.

Kind regards,

Haikel A. Lim, MD, MSc

Guest Editor

PLOS ONE

Additional Editor Comments (optional):

Thank you for your revised manuscript. I am pleased to convey that this manuscript is ready for publication in PLOS ONE. Thank you once again for your submission and professionalism throughout the review process. I wish you the best of luck in your future research endeavours.
---

## [Editor Report · Acceptance letter]

23 Jul 2020

PONE-D-20-01520R2 

Factors affecting anxiety among administrative officers working within the urgent protective action planning zone of a nuclear power station 

Dear Dr. Orita:

I'm pleased to inform you that your manuscript has been deemed suitable for publication in PLOS ONE. Congratulations! Your manuscript is now with our production department. 

Kind regards, 

on behalf of

Dr. Haikel A. Lim 

Guest Editor

PLOS ONE